# Multi-omics analysis and functional validation of CHEK1 as an independent prognostic biomarker in Pancreatic cancer

Xiaonan Wei[1], Ruirong Yan[2], Shanshan Wang[1], Yaru Jiang[3], Haibin Li[1], Yanping Li[1]*

**1** Precision Medicine Laboratory for Chronic Non-communicable Diseases of Shandong Province, Institute of Precision Medicine, Jining Medical University, Jining, Shandong, China, **2** Department of Respiratory and Critical Care Medicine, Guangzhou Medical University Third Affiliated Hospital, Guangzhou, Guangdong, China, **3** Key Laboratory of Endocrine Glucose & Lipids Metabolism and Brain Aging, Ministry of Education, Department of endocrinology, Shandong Provincial Hospital Affiliated to Shandong First Medical University, Jinan, Shandong, China

* yanpingli@mail.jnmc.edu.cn

## Abstract

### Background

Pancreatic cancer is a highly aggressive tumor with a poor prognosis due to challenging early diagnosis and limited treatment options. CHEK1, a crucial cell cycle regulator, is important in tumor development, but its role in pancreatic cancer remains under-researched in terms of expression, function, and regulation.

### Objective

To study the effect of CHEK1 on proliferation, migration and cell cycle of pancreatic cancer, and to construct a prognostic prediction model to investigate the effect of CHEK1 on the prognosis of pancreatic cancer.

### Methods

We conducted a systematic investigation into the expression characteristics and biological functions of CHEK1 in pancreatic cancer, utilizing an integration of bioinformatics analyses and in vitro experiments. Initially, we performed an in-depth analysis of CHEK1 expression profiles and their clinical significance in pancreatic cancer, drawing on data from several public databases, including UALCAN, TNMplot, and TISIDB. Subsequently, we validated the impact of CHEK1 on the malignant biological behaviors of pancreatic cancer cells, specifically focusing on proliferation and migration, through a series of in vitro cellular experiments.

### Results

The findings revealed that CHEK1 expression levels were significantly upregulated in pancreatic cancer tissues, which correlated positively with tumor pathological grade.

**Data availability statement:** All relevant data are within the manuscript and its Supporting Information files.

**Funding:** This research was supported by the National Natural Science Foundation of China ( No. 82002961, Yanping Li; No. 82101867, Shanshan Wang ). The funding agencies had no involvement in the research design, data acquisition, data analysis and interpretation, nor in the writing of the manuscript.

**Competing interests:** The authors declare no conflicts of interest related to this study.

**Abbreviations:** CHEK1, Checkpoint kinase 1; OS, Overall Survival; G2/M, Gap 2 phase/Mitosis phase; ROC, Receiver Operating Characteristic; AUC, Area Under roc Curve.

KEGG pathway enrichment analysis further indicated that CHEK1 exerts critical regulatory functions across multiple oncogenic pathways, including cellular proliferation, G2/M checkpoint control, DNA replication, and DNA damage repair. In vitro experimental demonstrated that CHEK1 overexpression substantially increased both the proliferative progression and migratory capacity of pancreatic cancer cells.

## Conclusions

Our study indicates that high CHEK1 expression could be an independent prognostic marker for pancreatic cancer and may drive cancer progression by influencing DNA replication and G2/M checkpoint pathways. These insights offer a foundation for future research and targeted precision therapy involving CHEK1.

## Introduction

Pancreatic cancer is characterized as a malignant neoplasm originating from the epithelial cells of the pancreatic ducts oracinar structures [1]. As reported by the International Agency for Research on Cancer (IARC), pancreatic cancer is the seventh most prevalent cancer globally and exhibits the fourth highest mortality rate among all cancers [2]. Moreover, projections indicate that by 2030, pancreatic ductal adenocarcinoma (PDAC) will become the second leading cause of cancer-related mortality in the United States [3]. The elevated mortality rate associated with pancreatic cancer can be attributed primarily to the absence of early symptoms, significant genetic heterogeneity, and pronounced resistance to treatment. Consequently, most patients receive a diagnosis at an advanced stage, thereby missing the optimal window for surgical intervention. This delay results in suboptimal treatment outcomes and contributes to a 5-year survival rate that remains below 10 percent [4,5]. With a better understanding of cancer cell molecular traits, there's an opportunity to develop targeted therapies to address the challenges posed by this lethal disease.

CHEK1 is an essential component of the CHEK family, serving as a serine/threonine-specific protein kinase that is integral to the regulation of cell cycle checkpoints [6]. It primarily coordinates DNA damage response mechanisms. In response to DNA damage, CHEK1 can initiate cell cycle arrest, thereby allowing a crucial time frame for DNA repair [7]. Research has demonstrated that DNA damage therapy can exert anticancer effects by inhibiting tumor cell proliferation and inducing cell cycle arrest [8]. In tumors deficient in the TP53 gene, cell cycle arrest mediated by checkpoint kinase 1 (CHEK1) is essential for tumor cell survival [9]. Consequently, targeting CHEK1 represents a promising therapeutic strategy, particularly for TP53-deficient cancers. Research indicates that CHEK1 is overexpressed in various tumors relative to adjacent normal tissues [10–13]. Tumor cells exhibiting elevated levels of CHEK1 acquire a survival advantage by resisting increased DNA damage [14]. Furthermore, CHEK1 expression is associated with tumor grade and disease recurrence, suggesting its significant role in tumorigenesis and progression [15,16]. Research has documented the occurrence of mutations in the CHEK1 gene accompanied by

microsatellite instability in endometrial, colorectal, and gastric cancers [17–20]. Beyond oncology, CHEK1 dysfunction has also been implicated in other diseases such as neurodegenerative disorders, where it is correlated with dementia and a diminished protective role in Alzheimer's disease [21]. While CHEK1 abnormalities have been extensively documented across various tumor types, their precise biological role in pancreatic cancer remains inadequately understood.

This study undertakes a comprehensive assessment of CHEK1 gene expression levels and their prognostic implications in pancreatic cancer, employing bioinformatics platforms such as Clinical Biosignal House (https://www.aclbi.com/static/index.html). This analysis is complemented by molecular biology experiments, including cellular clone formation assays, to thoroughly investigate the role of CHEK1 in pancreatic cancer and its underlying mechanisms (S1 Fig). The findings offer novel insights into the function of CHEK1 in pancreatic cancer and its influence on patient clinical outcomes.

## Materials and methods

### Analysis of CHEK1 gene expression at mRNA and protein levels

First, we analyzed the mRNA and protein expression levels of CHEK1 gene in pancreatic cancer tissues compared to normal pancreatic tissues using the UALCAN (http://ualcan.path.uab.edu/analysis.html) database. To validate these findings, we further examined CHEK1 mRNA expression using three additional independent databases including: GEPIA2 (http://gepia.cancer-pku.cn/index.html), TNMplot (https://tnmplot.com/analysis/) and the Clinical Bioinformatics Assistant (www.aclbi.com).

### Analyzing the relationship between CHEK1 expression and clinicopathological variables

Using the clinical information comparison function within the TCGA module of the Clinical Bioinformatics Platform (https://www.aclbi.com/static/index.html#/tcga), we compared differences in clinical and pathological characteristics between the CHEK1 high-expression group and the CHEK1 low-expression group in pancreatic cancer. Additionally, TISIDB (http://cis.hku.hk/TISIDB/index.php) is an online platform focused on tumor research that integrates various heterogeneous data types. GEPIA2(http://gepia.cancer-pku.cn/index.html) is an intuitive network application tool from the information of the TCGA and GTEX databases. We further utilized the TISIDB and GEPIA databases to analyze the association between CHEK1 expression and different stages and grades of pancreatic cancer.

### Examining the impact of the CHEK1 gene on pancreatic cancer prognosis

UALCAN is a comprehensive, user-friendly, and interactive web resource for analyzing cancer OMICS data. Therefore, using the TISIDB (http://cis.hku.hk/TISIDB/) and UALCAN (https://ualcan.path.uab.edu/) platforms, patients were stratified into high and low-expression groups according to the median CHEK1 expression level, and differences in overall survival between the two groups were compared. Subsequently, to evaluate the independent prognostic impact of CHEK1 expression and other clinical characteristics, we performed univariate and multivariate Cox proportional hazards regression analyses using the prognostic model (Nomogram) module on the Clinical Bioinformatics website (https://www.aclbi.com/static/index.html#/prognosis). Cox proportional hazards regression analysis was performed using the survival package, with results visualized via the "forestplot" package. Subsequently, based on significant variables, the "rms" package was used to construct survival plots predicting 1-, 3-, and 5-year overall recurrence rates. All core computations for the online website analyses were performed using R software (v4.0.3). Statistical significance was defined as $P < 0.05$.

### Analysis of CHEK1 gene expression in different pancreatic cancer cell lines

The expression of the CHEK1 gene across various pancreatic cancer cell lines was examined utilizing data from the Assistant for Clinical Bioinformatics database (www.aclbi.com). This analysis aimed to identify suitable cell lines for further investigation and validation. The expression matrix of the CHEK1 gene for pancreatic tumor cell lines was sourced from

the CCLE dataset (https://depmap.org/portal/data_page/?tab=allData). Statistical analyses were performed utilizing R software (version 4.0.3). A p-value of less than 0.05 was deemed indicative of statistical significance.

## Correlation analysis of the CHEK1 gene and related Pathways

Using the correlation analysis module of the Clinical Bioinformatics Home platform(https://www.aclbi.com/static/index.html#/functional_analysis), we investigated the association between CHEK1 gene expression levels and related pathways in 179 pancreatic cancer samples. The platform downloaded STAR-counts data and corresponding clinical information for pancreatic cancer from the TCGA database (https://portal.gdc.cancer.gov),extracted the TPM-formatted data, normalized it using log2(TPM + 1), and retained samples with both RNA-seq data and clinical information. We then performed analysis using the GSVA package in R software, selecting the parameter method = 'ssgsea' for single-sample gene set enrichment analysis (ssGSEA). Finally, Spearman's correlation analysis was employed to investigate the association between CHEK1 gene expression and pathway scores. Statistical analyses were performed using R software version 4.0.3. Results were considered statistically significant when the P-value was less than 0.05.

## Cell culture

The two pancreatic cancer cell lines used in this study were PANC-1 and MIA PaCa-2. PANC-1 cells were cultured in DMEM medium (HyClone, Logan, USA) supplemented with 10% fetal bovine serum (FBS; HyClone, Logan, USA). MIA-PACA-2 cells were cultured in DMEM medium (Hyclone, No.SH30243.01B) supplemented with 10% fetal bovine serum (FBS; HyClone, Logan, USA) and 2.5% horse serum (Procell, Wuhan,China).

## Vector construction and transfection

SiRNAs (siCHEK1 and negative control siRNA) were synthesized by Shanghai GenePharma Co. Ltd. (Shanghai, China); an empty vector control was used to control for general effects of siRNA expression. SiRNA sequences are shown in S1Table. pcDNA3.1-3xHA-C-CHEK1(NM_001274) overexpression plasmid was purchased from LST Bio-tech ShanDong Co, Ltd (Jinan,Shangdong, China). PANC-1 and MIA-PACA-2 cells were seeded into 6-well plates ($3 \times 10^5$/well), incubator at 37°C and 5% $CO_2$ for 24 h. siRNA transfection was divided into blank control group, NC-siRNA group and CHEK1 siRNA group in PANC-1 cells. Overexpression transfection was divided into blank control group, NC group and CHEK1 OE group. Transfection of the plasmids was conducted using jetPRIME Transfection Reagent (Polyplus, Illkirch, France) using manufacturer's recommendations.

## qRT-PCR

Total RNA was extracted with the Trizol reagent (Thermo Fisher Scientific). For cDNA synthesis and real-time PCR, SynScript® III RT SuperMix for qPCR (including gDNA Remover) and ArtiCanATM SYBR qPCR Mix were employed (TSINGKE, Beijing, China), utilizing the CFX96 Real-Time PCR Detection System (Bio-Rad, Hercules, CA, USA). All primers were designed and synthesized by TSINGKE. Primer sequences are shown in S1Table. Gene expression levels were quantified using the threshold cycle (Ct) method, with relative expression levels calculated via the 2–ΔΔCt method. β-actin served as the endogenous control for the mRNAs.

## Western blotting

Whole protein lysates were extracted from the specified cells utilizing RIPA lysis buffer (Thermo Fisher Scientific, USA) and subsequently denatured at 100°C for 15 minutes. The samples were then separated using Pre-SDS-PAGE (E-life Technology, China) and transferred onto PVDF membranes (Invitrogen, USA). The membranes were blocked with 5% skim milk and incubated overnight at 4°C with the appropriate primary antibody. This was followed by a 1-hour incubation at room temperature

with the secondary antibody. Antibody information is detailed in S2 Table. Detection and visualization was performed using the ECL luminescent solution (Beyotime, China), and the grayscale values were quantified using ImageJ software.

## Colony growth assay

Following seeding and adherence of the cells to 6-well plates at a density of 500 cells per well, the cells underwent the designated treatment for a duration of two weeks. Subsequently, the cells were fixed with 4% paraformaldehyde for 10 minutes and stained with Giemsa dye overnight. After a single rinse with distilled water, images were captured using a camera, and the colony numbers were quantified using GraphPad Prism 6 software.

## EdU assay

Pancreatic cancer cell lines with specific genes knocked down were seeded in 96-well plates and incubated for 48 hours. For the Edu staining assay, followed by the addition of 10μM Edu (5-ethynyl-2'-deoxyuridine) for an additional 2 hours. Subsequently, the cells were collected, and staining was conducted utilizing the BeyoClickTM Edu-594 cell proliferation assay kit (C0078S, Beyotime, Shanghai, China) in accordance with the manufacturer's instructions.

## Transwell

For transwell migration assays, we used 24-well inserts with an 8μm pore size polycarbonate membrane (Corning Costar, Lowell, MA, USA), 600 μL of media supplemented with 10% fetal bovine serum (FBS) was introduced into the lower chamber. Concurrently, cells resuspended in serum-free media were added to the upper insert following transfection. After 12 hours of incubation, the transwell membranes were fixed and stained with crystal violet. Subsequently, the cells adhering to the lower surface of the membrane were enumerated using a light microscope (Olympus, Tokyo, Japan) at a magnification of 200x.

## Cell cycle detection using flow cytometry

After transfection, cells were collected with trypsin and rinsed twice with ice-cold PBS, followed by 75% ethanol at 4 °C overnight. After centrifugation, the cells were mixed with the configured DNA staining solution (PI; 0.5 mg/mL; BioLegend, Logan, USA) and incubated at room temperature away from light for 15 min. Then, the stained cells were then analyzed by flow cytometry (CytoFLEX S, BECKMAN, USA).

As all these databases are publicly accessible and do not involve human or animal samples, the Ethics Committee of Jining Medical University has granted exemption from ethical review for this study.

## Statistical analysis

The experimental data were analyzed using GraphPad Prism statistical software. Group differences were assessed using either a Student's t-test or a one-way ANOVA, applying stringent statistical methods for comparative analysis. Data are presented as the mean ± standard deviation (SD). Statistical analyses were performed using R software, version 4.0.3. A p-value of less than 0.05 was considered indicative of statistical significance.

## Results

### Elevated CHEK1 levels in pancreatic cancer correlate with its pathologic grading

Initially, we conducted an investigation into the expression levels of CHEK1 using the UALCAN database. Our analysis demonstrated that CHEK1 expression was significantly elevated in pancreatic cancer compared to normal tissues, at both the mRNA and protein levels, as illustrated in Fig 1A and 1B. Furthermore, we compared the mRNA expression of CHEK1 in tumor samples and normal tissues utilizing the GEPIA2 (http://gepia2.cancer-pku.cn/#index), the Assistant for Clinical

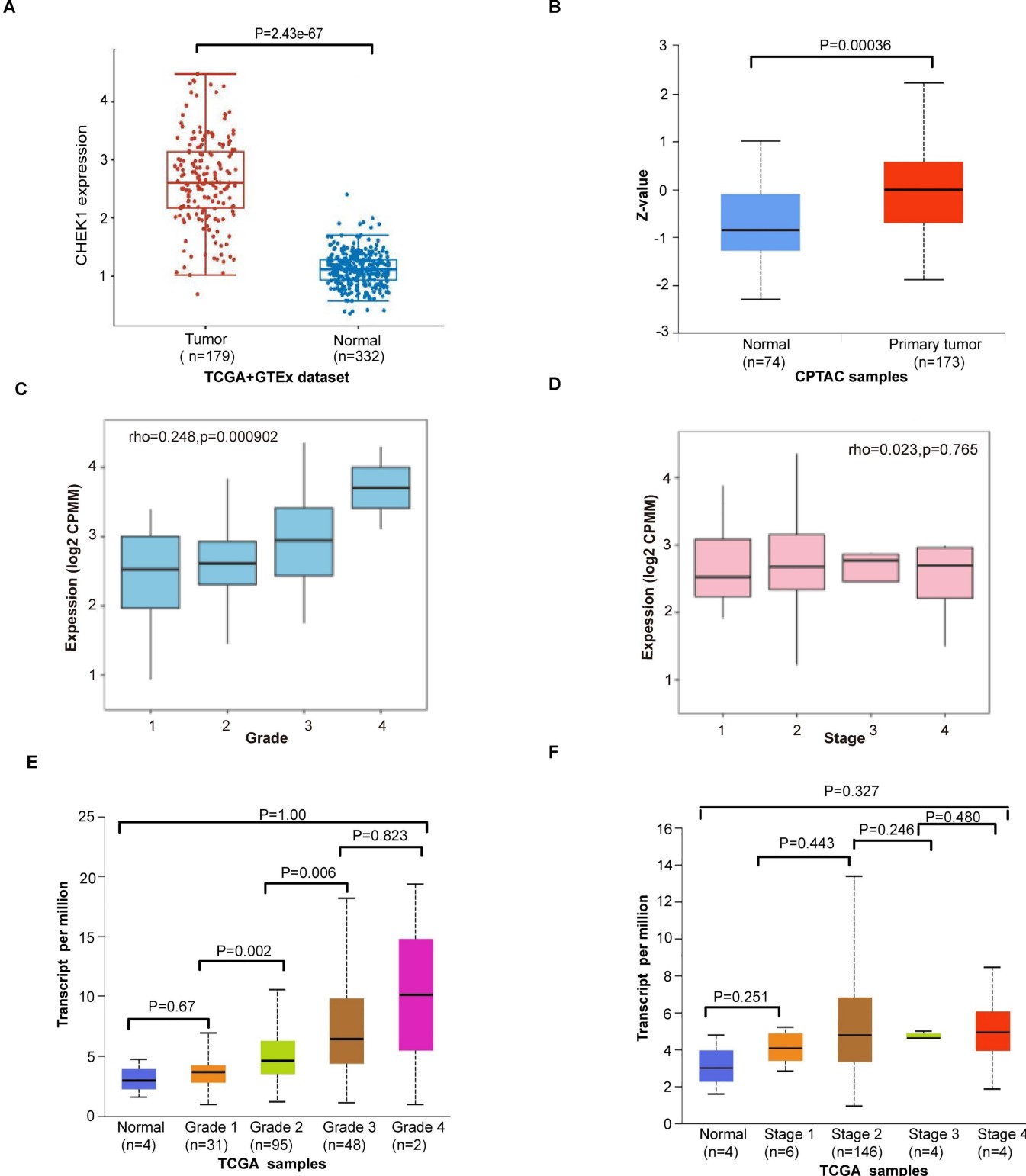

**Fig 1. Elevated CHEK1 expression is associated with grade in pancreatic cancer patients.** (A) Box plot showed the differential expression of CHEK1 in human normal tissues versus pancreatic cancer tissues within the TCGA database. (B) Box plot showed the differential protein levels of CHEK1 inhuman normal tissues and pancreatic cancer the by the UALCAN anslysis. (C) CHEK1 expression was analyzed in pancreatic cancers of

different grades via the TISIDAB website. (D) CHEK1 expression in pancreatic cancer at different stages was analyzed via the TISIDAB website. (E) Expression of CHEK1 in pancreatic cancer across different grades was compared and analyzed via the UALCAN website. (F) Expression of CHEK1 in pancreatic cancer across different stages was compared and analyzed via the UALCAN website.

Bioinformatics databases and TNMplot (https://tnmplot.com/analysis/) (S2A and S2B Fig). These databases corroborated our findings, indicating a pronounced upregulation of CHEK1 expression in pancreatic adenocarcinoma (PAAD) tissues relative to normal controls.

To further evaluate the clinical relevance of CHEK1, we analyzed its association with key clinicopathological characteristics in pancreatic cancer patients using the Clinical Bioinformatics Database (https://www.aclbi.com/static/index.html#/tcga). The analysis revealed that high CHEK1 expression was significantly associated with cancer grade but not with age, gender, race, pTNM stage, new tumor event type, or smoking (Table 1). Subsequently, we utilized the TISBID website (http://cis.hku.hk/TISIDB/index.php) and GEPIA2 (http://gepia2.cancer-pku.cn/#index) to examine the relationship between CHEK1 expression and the stage and grade of pancreatic cancers. The TISIDB database analysis shows CHEK1 expression levels are significantly positively correlated with grade (Fig 1C, p=0.000902) but not with stage (Fig 1D, p=0.765). The analysis conducted using the GEPIA database yielded a similar conclusion (Fig 1E and 1F). In

**Table 1.  Clinical characteristics of CHEK1 in Pancreatic cancer samples.**

| Classification | Characteristic | high expression | low expression | P-value |
|---|---|---|---|---|
| Status | Alive | 38 | 50 | |
| | Dead | 53 | 40 | 0.086 |
| Age | Mean (SD) | 65.4 (10.4) | 64 (11.4) | |
| | Median [MIN, MAX] | 66 [41,84] | 65 [35,88] | 0.377 |
| Gender | FEMALE | 40 | 40 | |
| | MALE | 51 | 50 | 1 |
| Race | ASIAN | 5 | 6 | |
| | BLACK | 2 | 4 | |
| | WHITE | 82 | 78 | 0.653 |
| pTNM_stage | IA | 2 | 3 | |
| | IB | 6 | 9 | |
| | IIA | 13 | 15 | |
| | IIB | 63 | 56 | |
| | III | 1 | 2 | |
| | IV | 4 | 3 | |
| | I | | 1 | 0.873 |
| Grade | G1 | 7 | 24 | |
| | G2 | 49 | 49 | |
| | G3 | 33 | 15 | |
| | G4 | 1 | 1 | |
| | GX | 1 | 1 | 0.003 |
| new_tumor_event_type | Metastasis | 37 | 23 | |
| | Metastasis:Recurrence | 1 | 1 | |
| | Primary | 1 | 1 | |
| | Recurrence | 10 | 10 | 0.811 |
| Smoking | Non-smoking | 30 | 36 | |
| | Smoking | 40 | 39 | 0.649 |

summary, CHEK1 exhibits elevated expression levels in pancreatic cancer and is positively correlated with the pathological grade of patients diagnosed with this malignancy.

## CHEK1 is an independent prognostic factor for pancreatic cancer

To examine the impact of CHEK1 gene expression on the prognosis of patients with pancreatic cancer. To investigate the influence of CHEK1 gene expression on the prognosis of pancreatic cancer patients, we conducted an analysis of Kaplan-Meier survival curves for overall survival (OS) using the TISIDB (http://cis.hku.hk/TISIDB/index.php) and UALCAN (https://ualcan.path.uab.edu/) online platforms. The analysis revealed a significant association between elevated CHEK1 expression and unfavorable prognosis in patients with pancreatic cancer (Fig 2A, p = 0.00523; Fig 2B, p < 0.0001). To further investigate the impact of the CHEK1 gene, along with clinical factors such as age, gender, and cancer stage on the prognosis of patients with pancreatic cancer, we conducted both single and multiple regression analyses of the CHEK1 gene utilizing the Assistant for Clinical Bioinformatics database (www.aclbi.com). The results of the univariate analysis, as depicted in Fig 2C, suggest that CHEK1 (Hazard Ratio, HR = 1.87578, p = 0.00003), age (HR = 1.02755, p = 0.00958), and grade (HR = 1.45334, p = 0.01028) significantly influence the prognosis of pancreatic cancer. Meanwhile, the impact of CHEK1 on prognosis remained statistically significant in the multivariate Cox regression analysis (Fig 2D, HR = 1.61139, p = 0.00349). The preceding analysis indicates that CHEK1 could potentially serve as an independent prognostic factor in pancreatic cancer. Therefore, we constructed a nomogram incorporating CHEK1, which was identified as significant in the multifactorial Cox regression analysis, to facilitate clinical prognosis. Fig 2E presents the nomograms for 1-year, 3-year, and 5-year overall survival (OS) within the cohort. The proximity of the nomogram model to the calibration curve, as depicted in Fig 2F, suggests that the model demonstrates a high predictive accuracy. Taken together, the analyses suggest that the CHEK1 gene is linked to survival in pancreatic cancer patients and can serve as a reliable independent prognostic factor.

## Identify suitable cell lines

To identify appropriate cell lines for subsequent functional experiments, we initially assessed the expression levels of CHEK1 across various pancreatic cancer cell lines utilizing the CCLE dataset. Our analysis revealed that Canpan-2 and PANC-1 exhibited the highest expression levels, whereas MIAPaCa-2 and SU.86.86 demonstrated the lowest (S3A Fig). We selected PANC-1 and MIAPaCa-2 cell lines for subsequent experiments. The expression of CHEK1 was confirmed at both the mRNA and protein levels in these cell lines, aligning with the findings from our network analysis (S3B and S3C Fig). Additionally, we constructed CHEK1 siRNA and overexpression (OE) expression vectors, which were validated through quantitative fluorescence PCR(S3D and S3E Fig) and Western blot analysis (S3F and S3G Fig). All raw data for quantitative real-time PCR and Western blot analyses are available in Supplementary Information, S1 Data.

## CHEK1 correlates with key oncogenic pathways in Pancreatic cancer

To further confirm the role of CHEK1 in pancreatic cancer, the correlations between CHEK1 gene and pathway score was analysed with Spearman by using online software (https://www.aclbi.com/static/index.html). The results showed that that CHEK1 expression in pancreatic cancer was significantly correlated with tumor proliferation (Fig 3A), G2M checkpoint (Fig 3B), DNA replication (Fig 3C) and DNA repair (Fig 3D).

In summary, integrated bioinformatics evidence from multiple databases collectively indicates that CHEK1 plays a crucial role in pancreatic cancer progression and patient prognosis.

## CHEK1 promotes PAAD cell progression

To elucidate the biological function of CHEK1 in pancreatic cancer cells, we established CHEK1-knockdown and overexpression models toassessed cellular phenotypes, focusing on proliferation and the cell cycle. Notably, CHEK1 depletion impaired colony formation in PANC-1 cells (Fig 4A), while its overexpression enhanced proliferation in MIA PaCa-2 cells (Fig 4B).

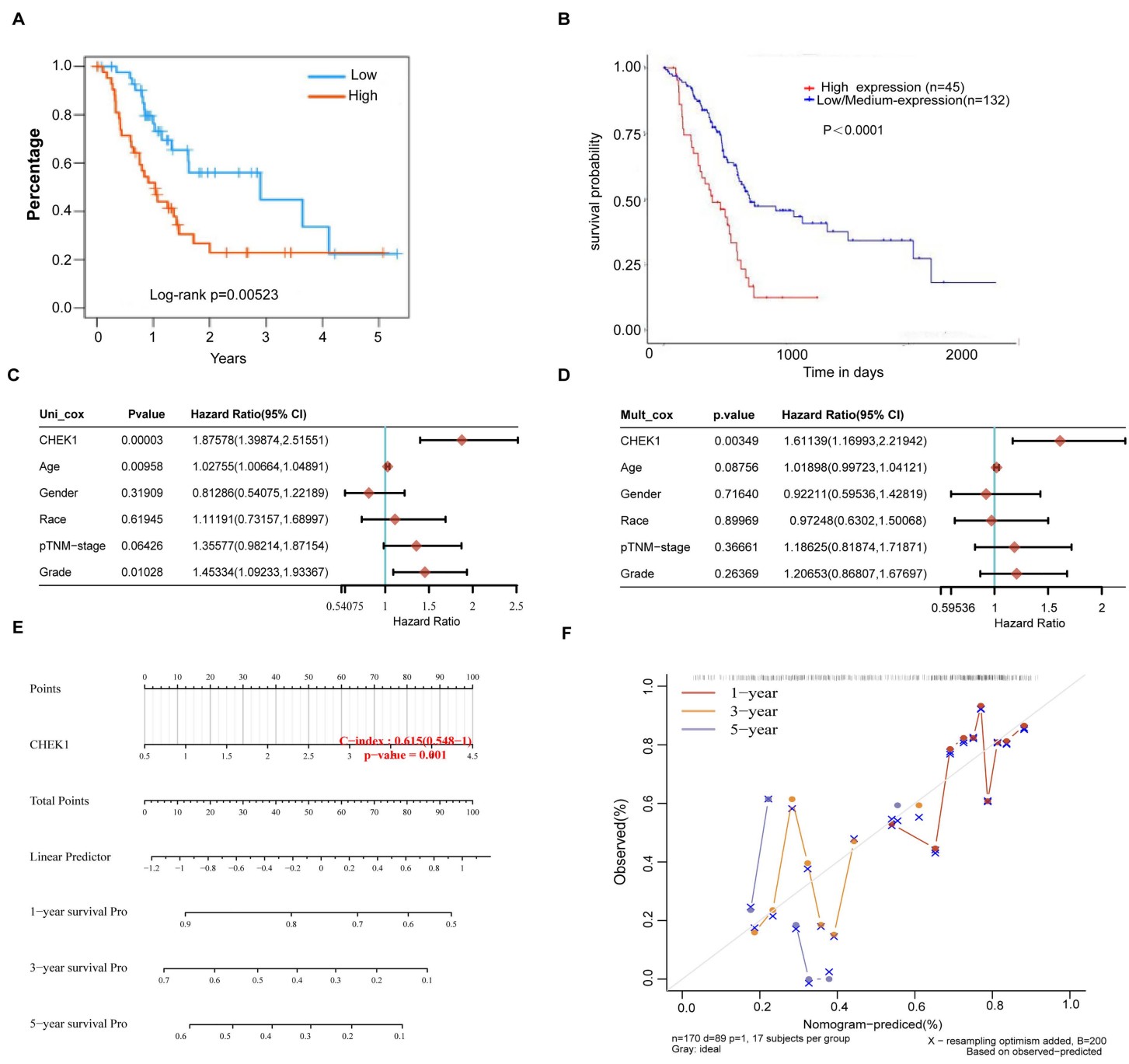

**Fig 2. An examination of the impact of the CHEK1 gene and various clinical factors on the prognosis of pancreatic cancer.** (A) overall survival of CHEK1 in TCGA-PAAD cohort from TISIDB online website. (B) overall survival of CHEK1 in TCGA-PAAD cohort from UALCAN online website. (C) A univariate Cox proportional hazards analysis was conducted to evaluate the p-values, risk coefficient (HR) and confidence interval associated with CHEK1 gene expression and clinical characteristics. (D) A multifactorial Cox proportional hazards regression analysis was employed to evaluate the p-values, risk coefficient (HR) and confidence interval corresponding to CHEK1 gene expression levels and various clinical characteristics. (E) The nomogram predicts the 1-year, 3-year, and 5-year overall survival rates of pancreatic cancer patients. (F) The calibration curve for the overall survival nomogram model in the discovery cohort is presented. The dashed diagonal line denotes the ideal nomogram, while the blue, red, and orange lines correspond to the observed nomogram predictions for 1-year, 3-year, and 5-year survival, respectively.

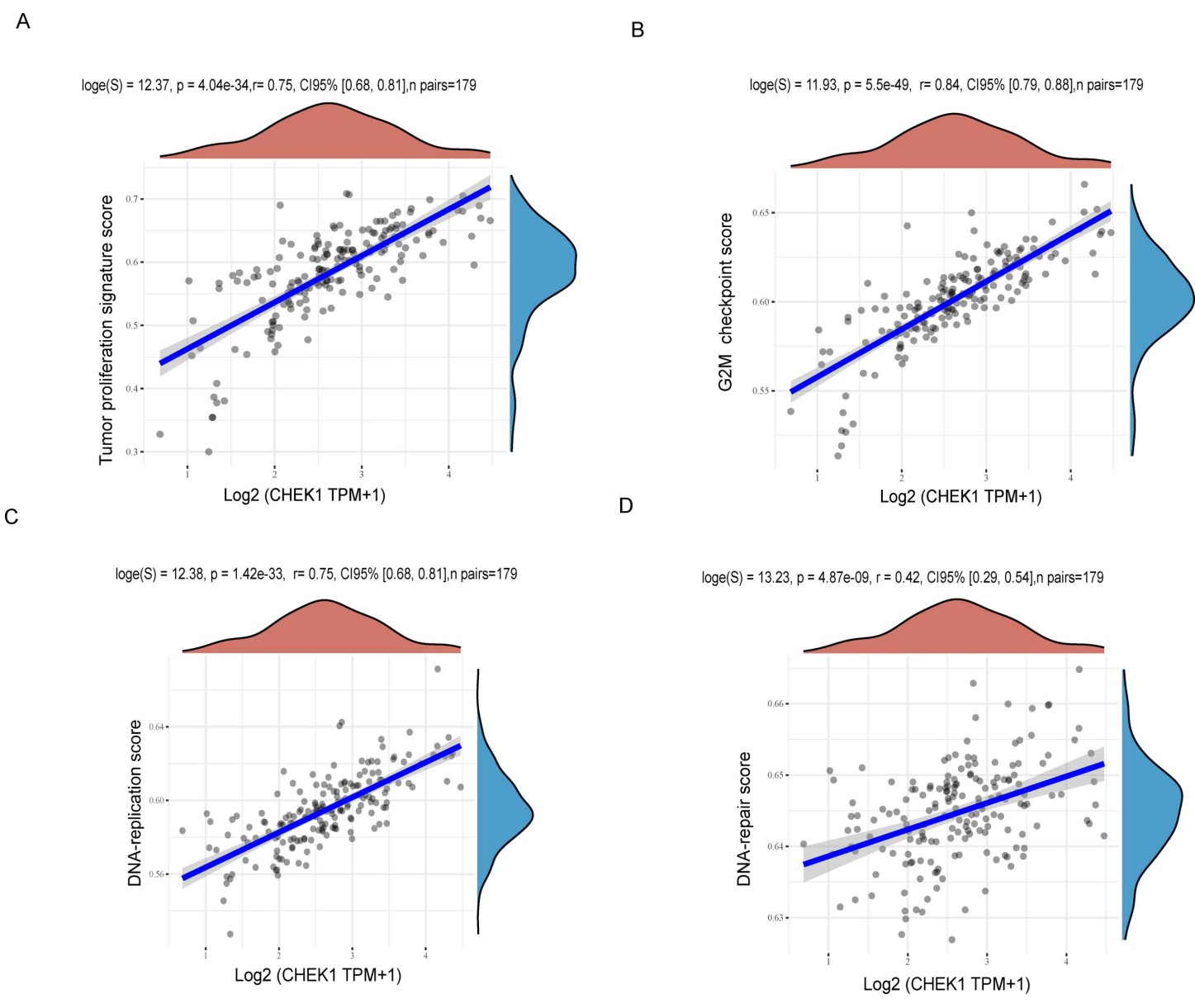

**Fig 3. The relationship between the CHEK1 gene and pathway score was examined using Spearman's correlation analysis.** The x-axis represents the distribution of gene expression, while the y-axis represents the distribution of the pathway score. (A) The correlations analysis between CHEK1 gene and tumor proliferation pathway. (B) The correlations analysis between CHEK1 gene and G2M checkpoint pathway. (C) The correlations analysis between CHEK1 gene and DNA-replication pathway. (D) The correlations analysis between CHEK1 gene and DNA repair pathway.

EdU staining provided additional confirmation that the knockdown of CHEK1 resulted in a reduced proportion of S-phase positive cell (Fig 4C), whereas the overexpression of CHEK1 led to an increased proportion of S-phase positive cells (Fig 4D). Subsequently, we used flow cytometry to assess the effect of CHEK1 on the cell cycle of pancreatic cancer. Following the knockdown of CHEK1, PANC-1 cells exhibited cell cycle arrest in the G1 phase, accompanied by a reduction in the proportion of cells in the S phase, decreased cellular synthesis and replication, and an overall deceleration of cell progression (Fig 5A). Conversely, the overexpression of CHEK1 resulted in cell cycle arrest in the S phase, an increased proportion of

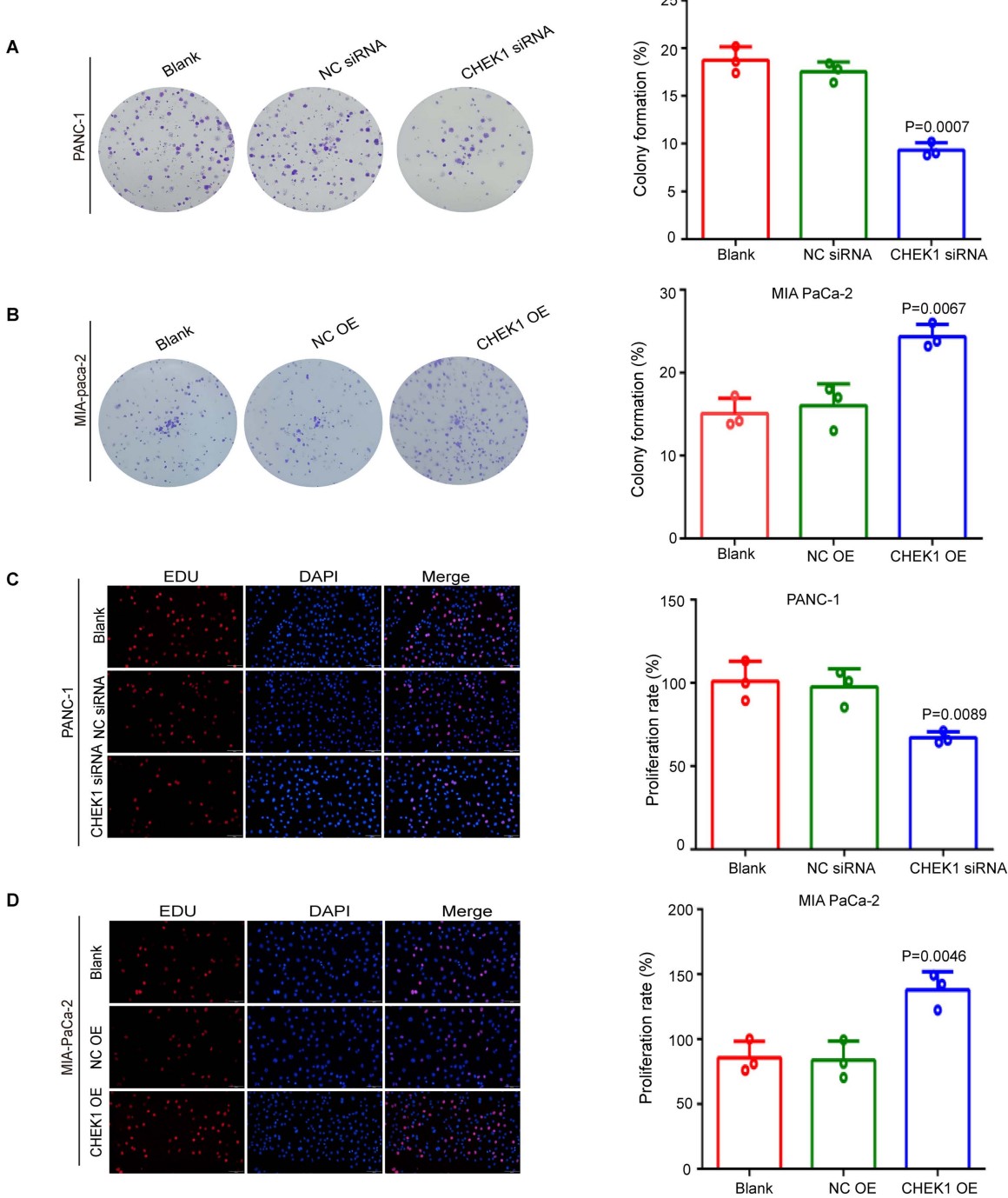

**Fig 4. Validation of CHEK1 function using colony formation and EdU assays.** (A) The effect of CHEK1 knockdown on the colony formation capability of PANC-1 cells (Crystal violet staining and statistical analysis). (B) The effect of overexpression of CHEK1 on the colony formation capability of MIAPaCa-2 cells (Crystal violet staining and statistical analysis). (C) Representative images of the EdU experiment,Scale bar: 50 μm. Statistical analysis of EdU experiments performed in PANC-1 cells. (D) Representative images of the EdU experiment, Scale bar: 50 μm. Statistical analysis of EdU experiments performed in MIAPaCa-2 cells.

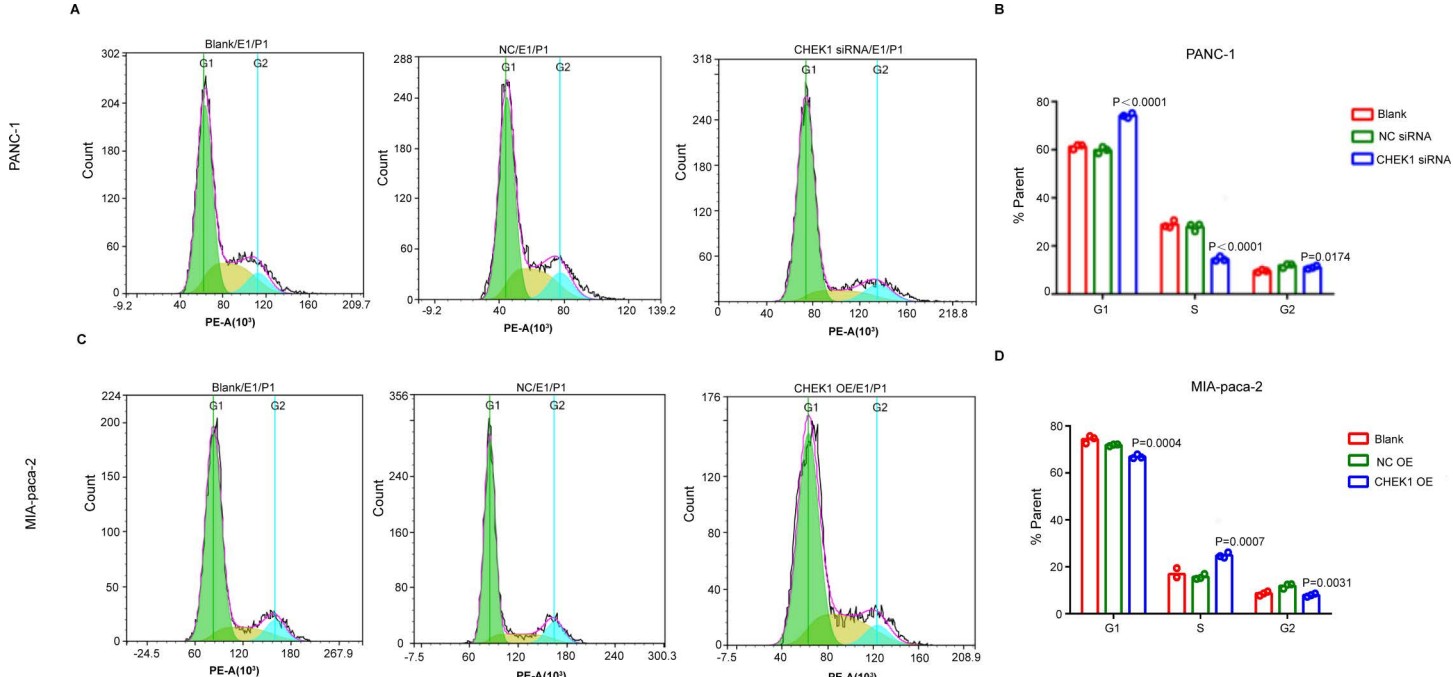

**Fig 5. CHEK1 influences the cell cycle in pancreatic cancer.** (A) Flow cytometric assessment of the impact of CHEK1 gene knockdown on cell cycle distribution in PANC-1 pancreatic cancer cells. (B) Statistical analysis of the impact of CHEK1 gene knockdown on the cell cycle distribution of PANC-1 pancreatic cancer cells. (C)Flow cytometric analysis of MIA-pcaca-2 cells following CHEK1 gene overexpression. (D) The cell cycle phase distribution in MIAPaCa-2 cells overexpressing CHEK1 was statistically analyzed.

cells in the S phase, enhanced cellular synthesis and replication, and facilitated progression of MIA PaCa-2 cells (Fig 5B). Together these data demonstrated that CHEK1 plays an important role in promoting pancreatic cancer growth.

## CHEK1 promotes PAAD cell invasion capacities

We conducted an in-depth analysis of the impact of CHEK1 on the migratory behavior of PANC-1 and MIA PaCa-2 pancreatic cancer cell lines using the transwell migration assay. The results indicated a significant reduction in the migration rates of PANC-1 cells in the CHEK1 siRNA-treated group compared to the Blank and Negative Control (NC) groups (Fig 6A). In contrast, MIA PaCa-2 cells exhibited a substantial increase in migration rates in the CHEK1 overexpression (CHEK1 OE) group compared to the Blank and NC groups (Fig 6B). Epithelial-mesenchymal transition (EMT) is a key process where epithelial cells gain mesenchymal traits, crucial for tumor invasion and metastasis. In this study, we conducted an in-depth analysis of the impact of CHEK1 knockdown and overexpression on the expression of epithelial-mesenchymal transition (EMT) markers. In cells with CHEK1 knockdown, there was an upregulation of the epithelial marker E-cadherin, accompanied by a downregulation of the mesenchymal markers N-cadherin and Vimentin (Fig 6C). Conversely, in cells with CHEK1 overexpression, E-cadherin expression was reduced, while N-cadherin and Vimentin expression levels were elevated (Fig 6D). These results indicate that CHEK1 facilitates the mesenchymal-epithelial transition (MET) and significantly promotes EMT in pancreatic cancer.

## Discussion

Recent advances in tumor molecular biology have highlighted the significance of cell cycle-regulated genes in cancer development. CHEK1, a key player in cell cycle checkpoints, is essential for genomic stability [22]. Despite numerous

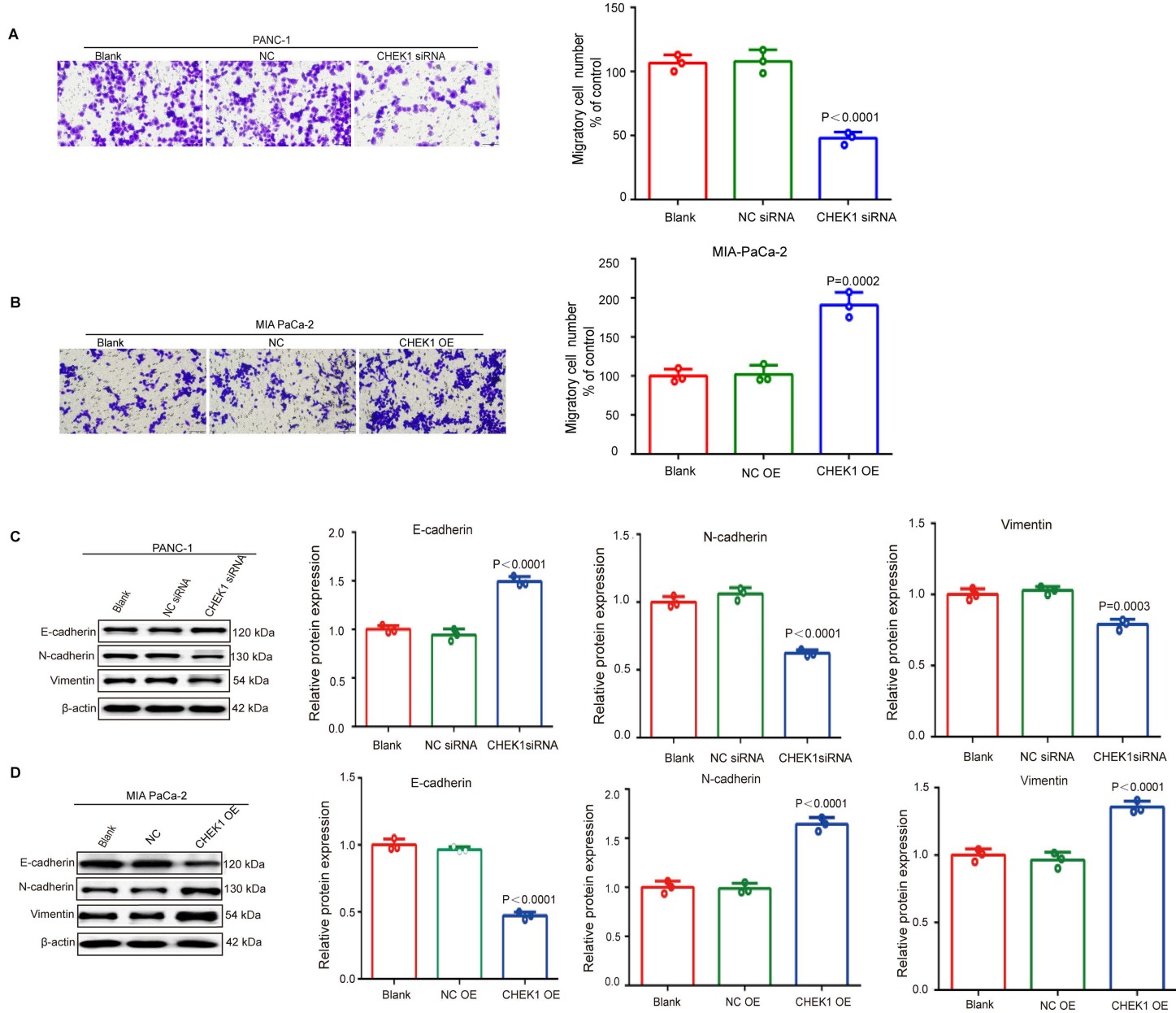

**Fig 6. The effect of the CHEK1 gene on the migratory ability of pancreatic cancer cells.** (A) Effect of knockdown of CHEK1 gene on PANC-1 cells migration of pancreatic cancer cells detected by Transwell assay (micrographs and quantitative analysis). (B) The effect of overexpression of CHEK1 gene on MIAPaCa-2 cells migration in pancreatic cancer cells was examined by Transwell assay (micrographs and quantitative analysis). (C) Changes in the expression levels of E-cadherin, N-cadherin and Vimentin after CHEK1 knockdown were analyzed by immunoblotting and ImageJ software. (D) Changes in the expression levels of E-cadherin, N-cadherin and Vimentin after overexpression of CHEK1 gene detected by immunoblotting and ImageJ software. Values represent the mean±SEM. ***P<0.001.

studies on CHEK1 in various cancers, its expression and prognostic role in pancreatic cancer remain underexplored. This study aims to elucidate CHEK1's role in pancreatic cancer using tools like multi-network and regression analyses, functional experiments, and flow cytometry analysis.

This study found that CHEK1 expression was significantly higher in pancreatic cancer tissues than in normal tissues andcorrelated positively with tumor grade. Consistent with this, elevated CHEK1 mRNA levels were linked to clinical stage [6,23]. Elevated CHEK1 protein levels were strongly associated with deep myometrial infiltration, suggesting CHEK1's potential role in endometrial cancer pathophysiology [24]. Univariate Cox regression analysis showed that high CHEK1 expression was significantly associated with poor overall survival (HR = 1.87578, p = 0.00003). Notably, this prognostic significance persisted in a multivariate Cox model after adjusting for key clinical confounders including age, sex, and tumor grade (HR = 1.61139, p = 0.00349). Thus, CHEK1 is an independent prognostic factor for pancreatic cancer, distinct from traditional clinicopathological parameters. This finding is strongly supported by existing literature across various cancers. For instance, a study screened and validated CHEK1 gene as significantly overexpressed in lung adenocarcinoma (LUAD) and lung squamous cell carcinoma (LUSC) by integrating multiple chip datasets from the GEO database, and its overexpression was significantly associated with poor patient prognosis [25]. Additional research using bioinformatics analysis and experimental validation has revealed that the CHEK1 gene is highly expressed in hepatocellular carcinoma and indicates poor prognosis. It promotes tumor proliferation and invasion by activating the transcription factor E2F1, making it a potential prognostic marker and therapeutic target [26]. In multiple myeloma, high CHEK1 expression is associated with poor outcomes, promoting chromosomal instability and drug resistance [27]. Collectively, these studies consolidate the role of CHEK1 as a biomarker for tumor progression and adverse prognosis. Functional experiments demonstrated that CHEK1 overexpression enhances pancreatic cancer cell growth, whereas its knockdown inhibits it, confirming its critical regulatory role. In hepatocellular carcinoma, CHEK1 activates CDC25C through phosphorylation, relieving CDK1 inhibition and promoting the G2/M phase transition, thus accelerating proliferation. This underscores CHEK1's crucial role in cell cycle regulation for these cancers [28]. To investigate its role in pancreatic cancer, we utilized the online platform (https://www.aclbi.com/static/index.html), Spearman correlation analysis revealed a significant association between CHEK1 expression and scores of the G2/M checkpoint and DNA replication pathways. Functionally, flow cytometry indicated that CHEK1 knockdown led to G1 phase arrest and slowed progression, while overexpression increased S phase cells, promoting replication and cell-cycle progression. Mechanistically, CHEK1 may promote tumor growth by modulating transcription factors like Myc to influence the expression of genes involved in cell proliferation [29].

Reduced E-cadherin expression is a key feature of epithelial-mesenchymal transition (EMT), which plays a major role in cancer invasion and metastasis. In many cancers, lower E-cadherin levels are often linked to enhanced migratory and invasive capabilities of tumor cells [30,31]. In non-small cell lung cancer, smoking induces epithelial-mesenchymal transition (EMT) by downregulating E-cadherin through histone deacetylase (HDAC), which is linked to poor prognosis in smokers [32]. N-cadherin and Vimentin serve as critical markers in the epithelial-mesenchymal transition (EMT) process [33,34]. In head and neck squamous cell carcinomas, increased N-cadherin and decreased E-cadherin expression mark the calcineurin transition, a key indicator of epithelial-mesenchymal transition (EMT) closely associated with tumor malignancy [35]. Previous studies in lung cancer have demonstrated that CHEK1 activates the Snail transcription factor, which suppresses E-cadherin and upregulates N-cadherin and Vimentin, thereby inducing EMT and promoting tumor cell migration and invasion [36]. In this study, we observed that the overexpression of CHEK1 led to an increase in both the transcriptional and translational levels of the N-cadherin and Vimentin genes within cells, thereby enhancing cellular migration. In contrast, the inhibition of CHEK1 was associated with a reduction in cell migration. These findings indicate that CHEK1 may contribute to the metastasis of pancreatic cancer by modulating the epithelial-mesenchymal transition (EMT) process.

## Conclusions

In summary, high CHEK1 expression in pancreatic cancer correlates with higher pathological grades and predicts poor outcomes. Functionally, CHEK1 enhances cancer cell proliferation, migration, and cell cycle progression, which makes it a key driver of cancer progression and a potential therapeutic target (Fig 7). Further research is needed to explore CHEK1-related molecular pathways to support targeted clinical interventions for pancreatic cancer.

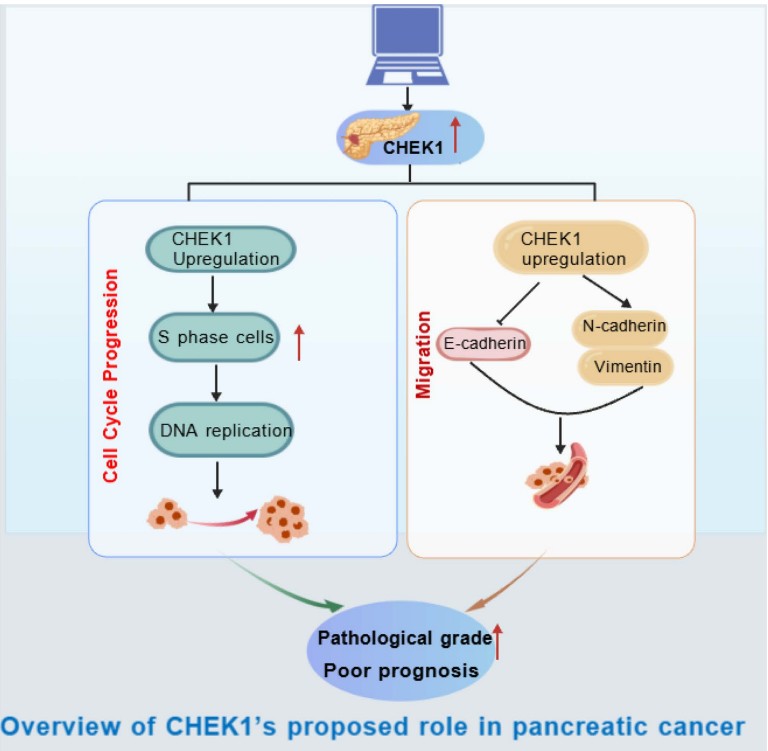

**Fig 7. Overview of CHEK1's propose role in pancreatic cancer.**

## Supporting information

**S1 Fig. Flowchart of Bioinformatics Analysis and Functional Validation of CHEK1 in Pancreatic Cancer.**
(TIF)

**S2 Fig. Differences in CHEK1 mRNA Expression Between Pancreatic Cancer and Normal Tissue.** (A) Analyze CHEK1 mRNA expression in pancreatic cancer and normal tissues using the GEPIA 2.0 website. (B) Analyze CHEK1 expression in pancreatic cancer tissue versus normal tissue using RNA sequencing data via the TNMplot online platform.
(TIF)

**S3 Fig. Analysis of CHEK1 expression and characterization in various pancreatic cancer cell lines.** (A) The bar chart illustrates the distribution of CHEK1 expression across various cell lines. In this chart, the horizontal axis denotes the gene expression status, while the vertical axis corresponds to the different cell lines. The height and color of the bars indicate the magnitude of gene expression, with the median value serving as the reference point for division. (B) Analysis of CHEK1 mRNA expression in PANC-1 and MIA PaCa-2 pancreatic cancer cell lines by quantitative real-time PCR (qRT-PCR). (C) Analysis of CHEK1 protein expression in PANC-1 and MIA PaCa-2 pancreatic cancer cell lines by Western blotting. (D) Validation of CHEK1 gene mRNA knockdown in the PANC-1 pancreatic cancer cell line by qRT-PCR. (E) Validation of the overexpression of CHEK1 gene mRNA levels in the pancreatic cancer cell line MIA PaCa-2 by qRT-PCR. (F) Validation of knockdown expression of the CHEK1 gene at the protein level in the PANC-1 pancreatic cancer cell line by Western blotting. (G) Validation of CHEK1 gene overexpression at the protein level in the MIA PaCa-2 pancreatic cancer cell line by Western blotting.
(TIF)

**S1 data. Raw quantitative data and analytical results from all functional experiments (including colony formation, EdU assay, cell migration, and cell cycle), along with uncropped, unadjusted original scanned images and labeling information from Western blot experiments.**
(PDF)

**S1 Table. List of primers for qRT-PCR and siRNA.**
(DOCX)

**S2 Table. List of antibodies for western blotting.**
(DOCX)

## Acknowledgments

We appreciate all the free online databases that have helped us with our research.

## Author contributions

**Data curation:** Ruirong Yan.

**Formal analysis:** Xiaonan Wei.

**Funding acquisition:** Yanping Li.

**Methodology:** Xiaonan Wei.

**Project administration:** Yaru Jiang, Haibin Li.

**Software:** Ruirong Yan, shanshan Wang.

**Supervision:** Ruirong Yan.

**Validation:** Xiaonan Wei.

**Writing – original draft:** Yanping Li.

**Writing – review & editing:** Yanping Li.

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
