## [Decision Letter · Decision Letter 0]

29 Sep 2025

Dear Dr. Li,

Thank you for submitting your manuscript to PLOS ONE. After careful consideration, we feel that it has merit but does not fully meet PLOS ONE’s publication criteria as it currently stands. Therefore, we invite you to submit a revised version of the manuscript that addresses the points raised during the review process.

Please make sure to adress all the concerns raised by the three experts in the field.

We look forward to receiving your revised manuscript.

Kind regards,

Hamidreza Montazeri Aliabadi

Academic Editor

PLOS ONE

Journal Requirements:

“This work was supported by the National Natural Science Foundation of China (No. 82002961, 82101867).”

4. In the online submission form you indicate that your data is not available for proprietary reasons and have provided a contact point for accessing this data. Please note that your current contact point is a co-author on this manuscript. According to our Data Policy, the contact point must not be an author on the manuscript and must be an institutional contact, ideally not an individual. Please revise your data statement to a non-author institutional point of contact, such as a data access or ethics committee, and send this to us via return email. Please also include contact information for the third party organization, and please include the full citation of where the data can be found.

6. Thank you for stating the following in your manuscript:

“This work was supported by the National Natural Science Foundation of China (No. 82002961, 82101867).”

“This work was supported by the National Natural Science Foundation of China (No. 82002961, 82101867).”

Reviewers' comments:

Reviewer's Responses to Questions

**Comments to the Author**

1. Is the manuscript technically sound, and do the data support the conclusions?

Reviewer #1: Yes

Reviewer #2: Yes

Reviewer #3: Partly

2. Has the statistical analysis been performed appropriately and rigorously?

Reviewer #1: Yes

Reviewer #2: No

Reviewer #3: I Don't Know

3. Have the authors made all data underlying the findings in their manuscript fully available?

Reviewer #1: Yes

Reviewer #2: No

Reviewer #3: Yes

4. Is the manuscript presented in an intelligible fashion and written in standard English?

Reviewer #1: Yes

Reviewer #2: Yes

Reviewer #3: Yes

Reviewer #1: accepted.

Please use the space provided to explain your answers to the questions above. You may also include additional comments for the author, including concerns about dual publication, research ethics, or publication ethics. (Please upload your review as an attachment if it exceeds 20,000 characters) (Limit 200 to 20000 Characters)

Reviewer #2: The authors of the manuscript PONE-D-25-40320 “Multi-omics Analysis and Functional Validation of CHEK1 as an Independent Prognostic Biomarker in Pancreatic Cancer” present a bioinformatic and experimental analysis of the role of CHEK1 in pancreatic cancer. They explore its involvement in specific cellular pathways and validate some findings through in vitro assays. Their bioinformatic results suggest that CHEK1 correlates with tumor pathological grade and is associated with key regulatory pathways. Overall, the manuscript provides supporting evidence for CHEK1 as a potential prognostic biomarker. While the study may be of interest to PLOS ONE readers, several issues must be addressed prior to acceptance.

Main Comments

Statistical Analysis, Experimental Design, Data Presentation, and Data Availability

These core areas require significant clarification and improvement. The manuscript would benefit greatly from better structuring of the methodology, clearer data presentation, and proper disclosure of dataset availability.

Figures/Schematic Representations

Consider including the following schematic illustrations to improve clarity and flow:

Figure 1: A diagram summarizing the bioinformatics pipeline, highlighting how data flowed into the functional validation experiments (including methods).

Figure 2: A conceptual model of CHEK1’s proposed role in pancreatic cancer, illustrating its potential as a biomarker. This figure could be included in Discussion.

Minor Comments

Cell Lines: Clearly explain the biological and molecular differences between MIA-PaCa-2 and PANC-1.

Line 85: Provide the web address for Clinical Biosignal House at its first mention.

Line 113: Clarify the comparison groups used for statistical analysis.

Lines 121–122, 129: These sentences are misplaced. Please move such information to the figure legends where appropriate.

Lines 131–132: Rewrite to include the names of the specific R packages used.

Lines 146–147: Provide at least one reference for the method used. Also clarify that “maestro” and “Seurat” are R libraries, and include the version numbers.

Lines 146–147 (continued): The t-SNE-based clustering and sub-clustering approach is poorly explained. What was the rationale? How was it implemented? Present and discuss the results accordingly.

Line 207: Indicate the incubation time for the relevant experiments.

Line 218: While the databases used are public, the specific datasets referenced in this study are not accessible. Please provide the repository link or accession numbers.

Lines 232–234: This sentence lacks context and appears disconnected. Please rewrite for clarity.

Line 259: Replace “univariable” with “univariate.”

Line 260: Define HR (Hazard Ratio) at first mention.

Lines 289–291: If “p Spearman” refers to the correlation coefficient, replace p with r. To avoid redundancy, report both r and p-values in Figure 3, and simplify the text to state that CHEK1 correlates significantly with tumor proliferation, G2M checkpoint, DNA replication, and DNA repair. Also, add “Score” to the y-axis labels.

siRNA Experiments: Please explain why siRNA treatment leads to reduced colony formation. Is this a direct effect on proliferation or another mechanism?

Line 318: Define EMT (epithelial–mesenchymal transition) at first mention.

Lines 345–346: The claim that multivariate analysis supports CHEK1 as an independent prognostic factor is vague. Please expand this section and include relevant literature references.

Statistical Testing:

Indicate the normality test used for each dataset.

Justify the use of Pearson vs. Spearman correlation analyses.

Cell Line Coverage: If siRNA and overexpression experiments were conducted in both PANC-1 and MIA-PaCa-2 cells, then all associated results (colony formation, cell cycle analysis, and migration assays) must be presented for both lines.

Figure-Specific Comments

Figure 1:

Add data points to all boxplots.

Replace asterisks with exact p-values.

Justify the use of z-values in Figure 1B, or standardize the y-axis across subpanels.

Figure 2B: Add a legend or label to the x-axis.

Figure 5: Replace barplots with boxplots including data points.

Figure 6:

Add p-values directly onto plots instead of using asterisks.

Replace barplots with boxplots showing individual data points.

Supplementary File S1: Maintain consistent plotting style throughout. I recommend using boxplots with individual data points for all relevant figures.

Reviewer #3: This manuscript describes a study on the checkpoint protein kinase CHEK1 and its expression in pancreatic cancer cells. The major conclusions reached by the authors are that higher CHEK1 expression correlates with higher tumor grade, and that CHEK1 expression drives cancer progression through its effect on cell proliferation, migration and on cell cycle progression. Evidence is also provided that the epithelial-mesenchymal transition is affected by CHEK1 expression. As a result, the authors suggest that CHEK1 could be a suitable target for pancreatic cancer therapy.

Expression of CHEK1 has been studied in several cancers, and so the studies presented here are not particularly novel. However, because pancreatic cancer is difficult to effectively treat, insights into the role of CHEK1 in pancreatic cancer progression are useful. Overall, the conclusions follow the data, but there are some issues that must be addressed.

1. In Fig. 1A, only 4 control samples were included. While the authors state that the result is significant, it is not clear what statistical test was used. To this point, asterisks are included in this and some other figures without explanation.

2. It is curious that the colony formation of PANC-1 and MIA-PaCa-2 cells is nearly identical (~16%, Figs 4 A & B, blank columns) given that the authors chose these two cell lines because of their differences in CHEK1 expression (Fig. S2). It would be useful for experiments to include siRNA and overexpression in both cell lines. Furthermore, a “normal” (non-cancer) pancreatic cell line should be tested to determine if the effects seen are cancer specific.

3. As the authors point out in their introduction, p53 status is an important factor in CHEK1 function. The two cell lines employed both have TP53 mutations. The authors have mined data on many pancreatic cancer cell lines (Fig. S2) and should include more cell lines in their various experimental procedures.

4. Although much of the manuscript is nicely written, there are several instances where editing is needed. For example, there are incomplete sentences and in one case, a paragraph that seems out of place (paragraph beginning on line 285). Thorough editing and proof-reading are required.

As indicated above, inclusion of additional experiments (more cell lines) would strengthen the arguments made in this manuscript and clarify details regarding the observed effects.

**Do you want your identity to be public for this peer review?** For information about this choice, including consent withdrawal, please see our Privacy Policy

Reviewer #1: No

Reviewer #2: No

Reviewer #3: No

---

## [Author Response · Author response to Decision Letter 1]

19 Nov 2025

Dear Editor:

We very much appreciated the reviewers’ constructive critiques and comments regarding our manuscript, which we have now completely revised according to the reviewers’ suggestions. We have added some additional content to address reviewers' concerns. Many critical changes have been made which are highlighted in red and included in this newly revised version. We offer here our point-by-point response to the reviewers’ questions and concerns:

Managing Journal Requirements:

Q1: Please ensure that your manuscript meets PLOS ONE's style requirements, including those for file naming. The PLOS ONE style templates can be found at

A1: We thank the editor for this reminder. We have carefully revised the manuscript to ensure it fully adheres to PLOS ONE's style requirements, in accordance with the journal's templates.

Q2: Please include your tables as part of your main manuscript and remove the individual files. Please note that supplementary tables (should remain/ be uploaded) as separate "supporting information" files.

A2: We are very grateful for the editor's important note. We have revised the manuscript accordingly: �1�The table in the main text (Table 1) should be placed directly within the main manuscript file in an editable format (Word table). �2�Supplementary tables (Table S1 and Table S2) have been uploaded as separate “supplementary information” files.

Q3: Please provide an amended statement that declares *all* the funding or sources of support (whether external or internal to your organization) received during this study, as detailed online in our guide for authors at http://journals.plos.org/plosone/s/submit-now. Please also include the statement “There was no additional external funding received for this study.” in your updated Funding Statement.

A3: Thank you for your guidance. We have amended the Funding Statement as required and included it in our cover letter.

Q4: In the online submission form you indicate that your data is not available for proprietary reasons and have provided a contact point for accessing this data. Please note that your current contact point is a co-author on this manuscript. According to our Data Policy, the contact point must not be an author on the manuscript and must be an institutional contact, ideally not an individual. Please revise your data statement to a non-author institutional point of contact, such as a data access or ethics committee, and send this to us via return email. Please also include contact information for the third party organization, and please include the full citation of where the data can be found.

A4:Thank you for raising this issue. We have revised the data availability statement as requested. The contact has been changed from the co-author to the Data Access Committee of the Precision Medicine Institute at Jining Medical University. The institution's contact information can be found in the Data availability statement section of the article.

Q5: PLOS ONE now requires that authors provide the original uncropped and unadjusted images underlying all blot or gel results reported in a submission’s figures or Supporting Information files. This policy and the journal’s other requirements for blot/gel reporting and figure preparation are described in detail at https://journals.plos.org/plosone/s/figures#loc-blot-and-gel-reporting-

requirements and https://journals.plos.org/plosone/s/figures#loc-preparing-figures-from-image-files. When you submit your revised manuscript, please ensure that your figures adhere fully to these guidelines and provide the original underlying images for all blot or gel data reported in your submission. See the following link for instructions on providing the original image data: https://journals.plos.org/plosone/s/figures#loc-original-images-for-blots-and-gels.

A5: Following the editor’s request�all original, uncropped, and unadjusted imprint images in this document have been uploaded as a Supporting Information file named ‘S1_raw_images’.

Q6: Thank you for stating the following in your manuscript:“This work was supported by the National Natural Science Foundation of China (No. 82002961, 82101867).”We note that you have provided additional information within the Acknowledgements Section that is not currently declared in your Funding Statement. Please note that funding information should not appear in the Acknowledgments section or other areas of your manuscript. We will only publish funding information present in the Funding Statement section of the online submission form.

“This work was supported by the National Natural Science Foundation of China (No. 82002961, 82101867).” Please include your amended statements within your cover letter; we will change the online submission form on your behalf.

A6: Following the editor's advice and in compliance with journal policy requirements, we have removed all references to funding sources from the manuscript and incorporated the revised statement into the cover letter.

Q7: If the reviewer comments include a recommendation to cite specific previously published works, please review and evaluate these publications to determine whether they are relevant and should be cited. There is no requirement to cite these works unless the editor has indicated otherwise.

A7: Thanks for your suggestion, we confirm that no specific citations were requested by the reviewers in their comments.

Reviewer #2:

Main Comments

Q1: Statistical Analysis, Experimental Design, Data Presentation, and Data Availability

These core areas require significant clarification and improvement. The manuscript would benefit greatly from better structuring of the methodology, clearer data presentation, and proper disclosure of dataset availability.

A1�We sincerely thank the reviewers for their critical and constructive comments on the core content of this manuscript. We have thoroughly revised the manuscript to address these issues, significantly enhancing the clarity, rigor, and reproducibility of the research. Key improvements are outlined below:

1�Statistical Analysis: In response to the reviewer's comment, we have added a new 'Statistical Analysis' subsection to the end of the 'Materials and Methods' chapter. This subsection provides a comprehensive and detailed exposition of statistical methods and highlighted in red to ensure all changes are clearly visible, line 265-271, page 9.

Q2: Figures/Schematic Representations

Consider including the following schematic illustrations to improve clarity and flow:

Figure 1: A diagram summarizing the bioinformatics pipeline, highlighting how data flowed into the functional validation experiments (including methods).

Figure 2: A conceptual model of CHEK1’s proposed role in pancreatic cancer, illustrating its potential as a biomarker. This figure could be included in Discussion.

A2 (1): We thank the reviewer for this suggestion. As recommended, we have now added a new diagram as Supplementary Figure 1 (S1 Fig.), which summarizes the bioinformatics pipeline and illustrates how the analysis led to the functional validation experiments.

A2 (2): We are grateful to the reviewer for this insightful comment. In response, we have developed a conceptual model that delineates the proposed mechanistic role of CHEK1 in PAAD and its potential clinical implications as a biomarker. This model is now presented as Fig 7 in the Conclusions section.

Minor Comments

Q1: Cell Lines: Clearly explain the biological and molecular differences between MIA-PaCa-2 and PANC-1.

A1: Thank you for your valuable suggestions. To clearly and concisely summarize the key differences, we have created a table comparing the fundamental characteristics of the MIA-PaCa-2 and PANC-1 cell lines.See

Table 1: Key Biological and Molecular Characteristics of the MIA-PaCa-2 and PANC-1 Human Pancreatic Cancer Cell Lines. Briefly, the two lines differ notably in their origin (primary pancreas tumor vs. metastatic site), key driver mutations (harboring different KRAS and TP53 alleles), and morphological appearance (epithelioid vs. more fibroblastic). We believe this table will provide readers with a quick and comprehensive reference for the distinct biological and molecular profiles of these commonly used models."

Feature MIA-PaCa-2 PANC-1 Primary Data Sources & Notes

Origin / Pathology Pancreatic carcinoma; Poorly differentiated Pancreatic ductal epithelium; Moderately differentiated ATCC Database; Deer et al., 2010

Key Genetic Alterations

KRAS mutation Homozygous G12C Heterozygous G12D ATCC,COSMIC,CCLE databases

TP53 status Mutated (R248W) Wild-type ATCC, COSMIC databases

CDKN2A/p16status Homozygously deleted Mutated or deleted Literature consensus

Molecular Subtype Basal-like/Quasi-mesenchymal Classical Collisson et al., 2011; Moffitt et al., 2015

Morphology attached epithelial with floating rounded cells epithelial ATCC Database

Invasive Potential (in vitro) Moderate High Deer et al., (2010)

From the perspective of functional tractability, as described in our manuscript (FigS2A), analysis of the CCLE database revealed that PANC-1 and MIA-PaCa-2 represent the two extremes of the CHEK1 expression spectrum. This “High-Low Contrast Model” strategy offers a critical advantage: knocking down CHEK1 in PANC-1 (high expression) allows observation of its loss-of-function phenotype; conversely, overexpressing CHEK1 in MIA-PaCa-2 (low expression) validates its gain-of-function effect. This ensures the robustness and universality of our findings. Second, this functional screening strategy is highly complementary to our previously outlined consideration of molecular heterogeneity. Despite differing baseline CHEK1 expression levels, these cell lines harbor distinct KRAS and TP53 mutation backgrounds. Thus, if CHEK1 demonstrates consistent key functions across such divergent genetic contexts, it significantly enhances the universality and clinical relevance of our findings. In summary, our selection is deliberate: the combination of PANC-1 and MIA-PaCa-2 not only provides an ideal functional comparison model but also enables us to assess the common role of CHEK1 across different molecular subtypes of pancreatic cancer.

Q2: Line 85: Provide the web address for Clinical Biosignal House at its first mention.

A2: We appreciate the reviewer's suggestions. As requested, we have now provided the official website of Clinical Biosignal House upon its first mention in the paper, See line 86, page 3.

Q3: Line 113: Clarify the comparison groups used for statistical analysis.

A3: We appreciate this important comment from the reviewers. We have made comprehensive revisions to the Materials and Methods section where line 113 appears, clearly defining the control group used for analysis. All modifications are highlighted in red in the revised manuscript. See line 126-136, page 5.

Q4: Lines 121-122, 129: These sentences are misplaced. Please move such information to the figure legends where appropriate.

A4: We appreciate the reviewers' suggestions. As recommended, the descriptive details in lines 121–122 and line 129 have been moved to the corresponding figure legends, and the Materials and Methods section has been comprehensively revised (see red markings, lines 153–166, page 6).

Q5: Lines 131-132: Rewrite to include the names of the specific R packages used.

A5: We thank the reviewer for this suggestion. We have revised the sentence in Lines 131-132 to specify the names and purposes of the key R packages used in our analysis, as requested.See line 158-162, page 6.

Q6: Lines 146-147: Provide at least one reference for the method used. Also clarify that “maestro” and “Seurat” are R libraries, and include the version numbers.

A6: We sincerely appreciate the reviewer's comments. However, we regret to inform you that upon verification, we did not obtain valid expression results for the CHEK1 gene in our pancreatic cancer tissue single-cell data. Since this result is not presented in the manuscript, to maintain consistency between the “Materials and Methods” and “Results” sections, we have decided to remove the corresponding analytical method description.

Q7: Lines 146-147 (continued): The t-SNE-based clustering and sub-clustering approach is poorly explained. What was the rationale? How was it implemented? Present and discuss the results accordingly.

A7: We appreciate the reviewers' valuable comments. Since no valid expression results for the CHEK1 gene were obtained in this dataset, we have decided to remove the descriptions of analytical methods mentioned by the reviewers—such as t-SNE clustering—from the manuscript to maintain consistency between the “Methods” and “Results” sections.

Q8: Line 207: Indicate the incubation time for the relevant experiments.

A8: We appreciate the reviewer's suggestions. The incubation time has been explicitly labeled as “after 12 hours of incubation.” This modification has been highlighted for the reviewer's reference,See line 250-251, page 9.

Q9: Line 218: While the databases used are public, the specific datasets referenced in this study are not accessible. Please provide the repository link or accession numbers.

A9:We appreciate the reviewer's important suggestion. All database URLs used in this study have been modified into accessible hyperlinks and are now displayed in blue.

Q10: Lines 232-234: This sentence lacks context and appears disconnected. Please rewrite for clarity.

A10: We sincerely appreciate the reviewer's suggestions. We have rewritten the relevant sections of the manuscript as advised to provide a more complete and coherent narrative. See line 287-290, page 10.

Q11: Line 259: Replace “univariable” with “univariate.”

A11: We thank the reviewer for this careful observation. The term has been corrected to "univariate" in the result,See line 315, page 11.

Q12: Line 260: Define HR (Hazard Ratio) at first mention.

A12: Thank you for this helpful suggestion. As recommended, we have defined "HR" as "Hazard Ratio" upon its first mention in the manuscript.See line 316, page 11.

Q13: Lines 289–291: If “p Spearman” refers to the correlation coefficient, replace p with r. To avoid redundancy, report both r and p-values in Figure 3, and simplify the text to state that CHEK1 correlates significantly with tumor proliferation, G2M checkpoint, DNA replication, and DNA repair. Also, add “Score” to the y-axis labels.

A13: We sincerely thank the reviewer for this valuable feedback. We have carefully revised the manuscript according to the suggestions, We have revised the manuscript by standardizing the correlation coefficient notation to "r", annotating Figure 3 with both r and p-values, simplifying redundant textual descriptions (See line 351-359, page 12), and clarifying y-axis labels with "Score" as suggested,See Fig 3.

Q14: siRNA Experiments: Please explain why siRNA treatment leads to reduced colony formation. Is this a direct effect on proliferation or another mechanism?

A14: We thank the reviewer for highlighting the important issue of reduced colony formation after CHEK1 knockdown. Based on our experimental data, siRNA-mediated knockdown of CHEK1 reduces clonogenicity through a process that directly affects cell proliferation. The core mechanism is as follows: CHEK1 deficiency induces G1 phase arrest, thereby reducing the proportion of cells in S phase. This directly inhibits DNA replication and cell division, ultimately weakening the cells' long-term proliferative capacity and cl

---

## [Decision Letter · Decision Letter 1]

8 Dec 2025

Dear Dr. Li,

Thank you for submitting your manuscript to PLOS ONE. After careful consideration, we feel that it has merit but does not fully meet PLOS ONE’s publication criteria as it currently stands. Therefore, we invite you to submit a revised version of the manuscript that addresses the points raised during the review process.

Please address all the reviewers' comments, and if possible, please consider seeking editorial help, as the reviewer 3 has recommended.

We look forward to receiving your revised manuscript.

Kind regards,

Hamidreza Montazeri Aliabadi

Academic Editor

PLOS One

Journal Requirements:

Reviewers' comments:

Reviewer's Responses to Questions

**Comments to the Author**

Reviewer #2: (No Response)

Reviewer #3: All comments have been addressed

2. Is the manuscript technically sound, and do the data support the conclusions?

Reviewer #2: Yes

Reviewer #3: Yes

3. Has the statistical analysis been performed appropriately and rigorously?

Reviewer #2: Yes

Reviewer #3: Yes

4. Have the authors made all data underlying the findings in their manuscript fully available?

Reviewer #2: No

Reviewer #3: Yes

5. Is the manuscript presented in an intelligible fashion and written in standard English?

Reviewer #2: Yes

Reviewer #3: No

Reviewer #2: The authors of the manuscript PONE-D-25-40320_R1 “Multi-omics Analysis and Functional Validation of CHEK1 as an Independent Prognostic Biomarker in Pancreatic Cancer” present a bioinformatic and experimental analysis of the role of CHEK1 in pancreatic cancer.

The manuscript has been significantly improved. However, some minor issues remain to be addressed. For instance, the authors indicate in their Response to Reviewers that Table 1 contains the “Key biological and molecular characteristics of the MIA-PaCa-2 and PANC-1 human pancreatic cancer cell lines,” but in the revised manuscript Table 1 is titled “Clinical characteristics of CHEK1 in pancreatic cancer samples.” Please provide the correct table.

I appreciate that web links have been provided; however, in the interest of transparency, I would like the datasets used for the statistical analyses (such as those described in lines 114–122) to be made available as supplementary material or uploaded to a public repository.

Please provide the N in the legend of those plots where datapoints were not depicted.

Finally, the manuscript still contains multiple spacing and formatting errors in the text; please revise it thoroughly.

Reviewer #3: The authors have addressed the original concerns, especially those regarding statistical tests and descriptions. A couple of recommended additional experiments were not conducted, but solid rationales for not pursuing them were provided. The issue that still remains is the quality of the English. There are many errors, too many to list here. It is recommended that the authors seek independent editorial help.

**Do you want your identity to be public for this peer review?** For information about this choice, including consent withdrawal, please see our Privacy Policy

Reviewer #2: No

Reviewer #3: No

---

## [Author Response · Author response to Decision Letter 2]

15 Dec 2025

Dear Editor,

We sincerely appreciate the thoughtful and constructive feedback from the reviewers on our manuscript. We have carefully revised the manuscript in accordance with all of the reviewers’ suggestions, and have incorporated additional content to address their concerns. Key revisions are highlighted in red within the newly revised version. Below, we provide a point-by-point response to each of the reviewers’ questions and comments.

Journal Requirements:

Q1: If the reviewer comments include a recommendation to cite specific previously published works, please review and evaluate these publications to determine whether they are relevant and should be cited. There is no requirement to cite these works unless the editor has indicated otherwise.

A1: In response to the journal's requirements, we confirm that all necessary checks have been completed. Notably, the reviewers did not suggest any changes to the references. We have nevertheless thoroughly verified the reference list for completeness, accuracy, and proper formatting, and confirmed that no retracted articles are cited. All relevant updates are reflected in the revised manuscript.

Reviewer #2:

Q1: The manuscript has been significantly improved. However, some minor issues remain to be addressed. For instance, the authors indicate in their Response to Reviewers that Table 1 contains the “Key biological and molecular characteristics of the MIA-PaCa-2 and PANC-1 human pancreatic cancer cell lines,” but in the revised manuscript Table 1 is titled “Clinical characteristics of CHEK1 in pancreatic cancer samples.” Please provide the correct table.

A1: We thank the reviewer for carefully pointing out the confusion regarding the table numbering. We sincerely apologize, as this likely resulted from insufficient clarity in our description within the Response to Reviewers.

The "Table 1" referenced in our response letter is a supplementary table specifically prepared to address your earlier comment regarding "clarifying the biological and molecular differences between the two cell lines." Its content details the "Key biological and molecular characteristics of the MIA-PaCa-2 and PANC-1 human pancreatic cancer cell lines." This table was intended solely to directly respond to your inquiry and was not planned for inclusion in the main manuscript text.Change "See Table 1" to "Please see the table below."

The "Table 1" in the main manuscript is the pre-existing table from the original submission that analyzes clinical data, titled "Clinical characteristics of CHEK1 in pancreatic cancer samples."

Therefore, the response to the first issue raised in the reviewer's minor comments has been revised.

Minor Comments:

Q1: Cell Lines: Clearly explain the biological and molecular differences between MIA-PaCa-2 and PANC-1.

A1: Thank you for your valuable suggestions. To clearly and concisely summarize the key differences, we have created a table comparing the fundamental characteristics of the MIA-PaCa-2 and PANC-1 cell lines. Please see the table below: Key Biological and Molecular Characteristics of the MIA-PaCa-2 and PANC-1 Human Pancreatic Cancer Cell Lines. Briefly, the two lines differ notably in their origin (primary pancreas tumor vs. metastatic site), key driver mutations (harboring different KRAS and TP53 alleles), and morphological appearance (epithelioid vs. more fibroblastic). We believe this table will provide readers with a quick and comprehensive reference for the distinct biological and molecular profiles of these commonly used models."From the perspective of functional tractability, as described in our manuscript (FigS2A), analysis of the CCLE database revealed that PANC-1 and MIA-PaCa-2 represent the two extremes of the CHEK1 expression spectrum. This “High-Low Contrast Model” strategy offers a critical advantage: knocking down CHEK1 in PANC-1 (high expression) allows observation of its loss-of-function phenotype; conversely, overexpressing CHEK1 in MIA-PaCa-2 (low expression) validates its gain-of-function effect. This ensures the robustness and universality of our findings. Second, this functional screening strategy is highly complementary to our previously outlined consideration of molecular heterogeneity. Despite differing baseline CHEK1 expression levels, these cell lines harbor distinct KRAS and TP53 mutation backgrounds. Thus, if CHEK1 demonstrates consistent key functions across such divergent genetic contexts, it significantly enhances the universality and clinical relevance of our findings. In summary, our selection is deliberate: the combination of PANC-1 and MIA-PaCa-2 not only provides an ideal functional comparison model but also enables us to assess the common role of CHEK1 across different molecular subtypes of pancreatic cancer.

Q2: I appreciate that web links have been provided; however, in the interest of transparency, I would like the datasets used for the statistical analyses (such as those described in lines 114–122) to be made available as supplementary material or uploaded to a public repository.

A2: We thank you for your suggestion and sincerely apologize for any misunderstanding caused by the lack of clarity in our previous description.In response to your comments, we have revised the content in lines 114–122 to clarify that "all analyses were conducted using the standardized modules of the specified online website."See red line 125-138, page 5. Additionally, the correlation analysis of the CHEK1 gene and its associated pathways in the Materials and Methods section has been revised. See red lines 150–153 on page 5 and red lines 161–162 on page 6.

The public data underwent normalization processing on the platform, generating formatted data ready for direct analysis. All data used in this study can be accessed through the fixed analysis modules of the website, as described in the manuscript.To meet the reviewers' requirements while preserving the article's structure, we have downloaded the raw data used for analysis on the platform for the reviewers' review. This data is provided in separate files named “Raw Data for Pathway Correlation Analysis” and “Raw Data for Prognostic Model Analysis.”

Q3�Please provide the N in the legend of those plots where datapoints were not depicted.

A3: We appreciate the valuable suggestions provided by the reviewers. In accordance with their requests, we have thoroughly reviewed all figures that do not directly display data points and confirmed that only Figures 1C and 1D lack explicit labeling of data points. Regarding the suggestion for supplemental sample size information, we have cross-referenced the original data sources and relevant literature and confirmed that the currently cited data does not provide specific sample size details. We have also actively sought alternative publicly available data platforms to meet the reviewers' requirements but have not yet identified a suitable data source that simultaneously covers both Stage and Grade analyses. The relevant references are listed below for the reviewers' further consideration.

References: Ru B, Wong CN, Tong Y, Zhong JY, Zhong SSW, Wu WC, Chu KC, Wong CY, Lau CY, Chen I, Chan NW, Zhang J. TISIDB: an integrated repository portal for tumor-immune system interactions. Bioinformatics. 2019 Oct 15;35(20):4200-4202. doi: 10.1093/bioinformatics/btz210. PMID: 30903160.

Q4: Finally, the manuscript still contains multiple spacing and formatting errors in the text; please revise it thoroughly.

A4: Thank you for this important reminder. We sincerely apologize for the spacing and formatting errors that remained in the previous version. We have now conducted a thorough, line-by-line revision of the entire manuscript to correct these issues and have also polished the language. Please refer to the 'Revised Manuscript with Track Changes' document for details.

Reviewer #3:

Q1: The authors have addressed the original concerns, especially those regarding statistical tests and descriptions. A couple of recommended additional experiments were not conducted, but solid rationales for not pursuing them were provided. The issue that still remains is the quality of the English. There are many errors, too many to list here. It is recommended that the authors seek independent editorial help.

A1: We sincerely thank the reviewer for their valuable feedback. In accordance with the suggestions, the entire manuscript has undergone language polishing and revision, with all changes highlighted in red. Please refer to the 'Revised Manuscript with Track Changes' document for details.

---

## [Decision Letter · Decision Letter 2]

29 Dec 2025

Multi-omics analysis and functional validation of CHEK1 as an independent prognostic biomarker in Pancreatic cancer

PONE-D-25-40320R2

Dear Dr. Li,

We’re pleased to inform you that your manuscript has been judged scientifically suitable for publication and will be formally accepted for publication once it meets all outstanding technical requirements.

Kind regards,

Hamidreza Montazeri Aliabadi

Academic Editor

PLOS One

Additional Editor Comments (optional):

Reviewers' comments:

Reviewer's Responses to Questions

**Comments to the Author**

Reviewer #2: All comments have been addressed

Reviewer #3: All comments have been addressed

2. Is the manuscript technically sound, and do the data support the conclusions?

Reviewer #2: Yes

Reviewer #3: Yes

3. Has the statistical analysis been performed appropriately and rigorously?

Reviewer #2: Yes

Reviewer #3: Yes

4. Have the authors made all data underlying the findings in their manuscript fully available?

Reviewer #2: Yes

Reviewer #3: Yes

5. Is the manuscript presented in an intelligible fashion and written in standard English?

Reviewer #2: Yes

Reviewer #3: Yes

Reviewer #2: The authors addressed satisfactorily all questions and comments. This manuscript is now acceptable for publication in PLoSOne.

Reviewer #3: (No Response)

**Do you want your identity to be public for this peer review?** For information about this choice, including consent withdrawal, please see our Privacy Policy

Reviewer #2: No

Reviewer #3: No

---

## [Editor Report · Acceptance letter]

PONE-D-25-40320R2

PLOS One

Dear Dr. Li,

I'm pleased to inform you that your manuscript has been deemed suitable for publication in PLOS One. Congratulations! Your manuscript is now being handed over to our production team.

Kind regards,

on behalf of

Dr. Hamidreza Montazeri Aliabadi

Academic Editor

PLOS One